# Isoflurane mediated neuropathological and cognitive impairments in the triple transgenic Alzheimer's mouse model are associated with hippocampal synaptic deficits in an age-dependent manner

**Donald J. Joseph**[1][⊙]**, Chunxia Liu**[1,2][⊙]**, Jun Peng**[1,3]**, Ge Liang**[1]**, Huafeng Wei** [1]*

**1** Department of Anesthesiology and Critical Care, Perelman School of Medicine, University of Pennsylvania, Philadelphia, PA, United States of America, **2** Department of Anesthesiology, China-Japan Friendship Hospital, Beijing, China, **3** Department of Anesthesiology, sun Yat-sen Memorial Hospital, Sun Yat-Sen University, Guangzhou, China

⊙ These authors contributed equally to this work.
* Huafeng.wei@uphs.upenn.edu

**Data Availability Statement:** All relevant data are within the manuscript and its Supporting Information files.

## Abstract

Many *in vivo* studies suggest that inhalational anesthetics can accelerate or prevent the progression of neuropathology and cognitive impairments in Alzheimer Disease (AD), but the synaptic mechanisms mediating these ambiguous effects are unclear. Here, we show that repeated exposures of neonatal and old triple transgenic AD (3xTg) and non-transgenic (NonTg) mice to isoflurane (Iso) distinctly increased neurodegeneration as measured by S100β levels, intracellular Aβ, Tau oligomerization, and apoptotic markers. Spatial cognition measured by reference and working memory testing in the Morris Water Maze (MWM) were altered in young NonTg and 3xTg. Field recordings in the cornu ammonis 1 (CA1) hippocampus showed that neonatal control 3xTg mice exhibited hypo-excitable synaptic transmission, reduced paired-pulse facilitation (PPF), and normal long-term potentiation (LTP) compared to NonTg controls. By contrast, the old control 3xTg mice exhibited hyper-excitable synaptic transmission, enhanced PPF, and unstable LTP compared to NonTg controls. Repeated Iso exposures reduced synaptic transmission and PPF in neonatal NonTg and old 3xTg mice. LTP was normalized in old 3xTg mice, but reduced in neonates. By contrast, LTP was reduced in old but not neonatal NonTg mice. Our results indicate that Iso-mediated neuropathologic and cognitive defects in AD mice are associated with synaptic pathologies in an age-dependent manner. Based on these findings, the extent of this association with age and, possibly, treatment paradigms warrant further study.

**Funding:** This work was supported by National Institute of General Medical Science, National Institute of Health, Baltimore, Maryland, R01 NIH to HW (K08-GM073224, R01GM084979, 3R01GM084979-02S1, 2R01GM084979-06A1), and March of Dimes Birth Defects Foundation Research Grant (#12-FY08-167 to H.W.), White Plains, New York, and the bridging fund from the Department of Anaesthesiology, Perelman School of Medicine, University of Pennsylvania.

**Competing interests:** The authors have declared that no competing interests exist.

**Abbreviations:** 3xTg, Triple-transgenic; AD, Alzheimer's disease; fEPSP, field excitatory postsynaptic potential; MWM, Morris water maze; NonTg, Non-transgenic; LTP, Long term potentiation; Iso, isoflurane; ER, Endoplasmic reticulum; MAC, minimum alveolar concentration; ELISA, enzyme-linked immunosorbent assay; ABC, Avidin Biotin horseradish peroxidase Complex; TBS, Tris-buffered saline; PBS, Phosphate buffered saline; SDS, Sodium dodecyl sulfate; BSA, Bovine serum albumin; Bcl-2, B-cell lymphoma 2; aCSF, Artificial cerebral spinal fluid; SC, Schaffer collateral; CA1, cornu ammonis 1; HFS, high frequency stimulation; I/O, Input-output; ANOVA, one-way analysis of variance; PPR, Paired-pulse ratio; PPI, paired-pulse interval; PPF, paired-pulse facilitation; SR, stratum radiatum; Aβ, Amyloid beta; PND, Postnatal day; STP, short term plasticity; NMDA, N-methyl-D-aspartate.

# Introduction

Recent epidemiological evidence indicates that life experiences, including surgeries and multiple exposures to general anesthetics, are associated with AD [1–5]. Given that AD has emerged primarily as an affliction of the aging population [6] and the increasing incidence of anesthetic exposures with aging [7, 8], there has been significant interest in the pathologic mechanisms by which inhalational anesthetics alter the progression and pathogenesis of AD.

Indeed, emerging evidence from many AD mouse models suggests that general anesthetics impinge on neuropathology and cognitive functions[9]. Notably, many *in vivo* findings suggest that exposure to general anesthetics might exacerbate neuropathology in AD mice [10, 11]. However contradictory results in which AD mice undergoing single or repeated exposures to inhalational anesthetics with no immediate or long-lasting enhancement in neuropathology have been described as well [10, 11]. The effects of general anesthetics on cognition are also ambiguous, with inhalational anesthetics appear capable of improving cognition while also capable of exacerbating and mitigate the progression of its impairments in AD mice [9, 12–15].

The noted ambiguity in anesthetics mediated effects on cellular pathology and cognition likely reflects the differences in exposure paradigms, age, and experimental approaches. Nonetheless, these results suggest that anesthetics can induce complex cellular and behavioral changes in AD mice later in life when compared to age-matched non-transgenic mice, but the relationship between these effects and synaptic efficacy has not been studied. Given that the regulation of synaptic transmission is a fundamental property of neural circuits and synaptic loss is one of the best correlates of cognitive deficits in human[16], we investigated the relationship between anesthetics-mediated effects on cellular/cognitive pathology and synaptic functions in pre- and post-symptomatic AD mice in order to simultaneously define the biological processes disrupted by anesthetics and to understand the resulting functional abnormalities manifested later in life. Our results show that repeated exposures of neonatal and old mice to Iso distinctly altered histopathological markers and synaptic properties in 3xTg and NonTg mice. Specifically, the histopathological AD markers 6E10 and AT180 were significantly increased in neonates and old 3xTg mice, respectively. As expected, Iso impinged on neurodegeneration only in neonates by increasing the apoptotic markers Bcl-2 and Caspase 9 selectively in 3xTg mice and Caspase 12 along with the neurodegenerative S100β only in Non-Tg mice. The histopathological deficits correlated with impaired reference learning and working memory in the MWM test in 3xTg mice exposed as neonates, but those measures were not affected in age-matched NonTg mice. Although Iso exposures had no measureable impact on histopathology in old NonTg mice, long-term working memory was surprisingly improved in those mice. The noted histopathological and behavioral changes in NonTg and AD mice were associated with distinct synaptic deficits. Notably, repeated Iso exposures reduced the slope amplitude of LTP in 3xTg mice treated as neonates, but normalized AD-related LTP deficits in old 3xTg mice. In spite of the normalized LTP, Iso depressed basal synaptic transmission and PPF of synaptic release in these old 3xTg mice. In contrast to 3xTg mice, NonTg mice treated with Iso as neonates displayed reduced basal synaptic transmission and PPF, whereas Iso depressed only LTP in the old NonTg mice. Our results indicate that rather than exacerbating or improving histopathologic, cognitive, and synaptic measures in 3xTg mice relative to NonTg mice, repeated Iso exposures distinctively impinged on different aspects of those measures in age and genotype specific manners.

# Materials and methods

## Transgenic mice

All procedures were carried out in accordance with protocols approved by the Institutional Animal Care and Use Committee at the University of Pennsylvania. A total of 160 Non-

transgenic (C57BL6, RRID: MGI:5656552) and homozygote triple transgenic Alzheimer (3xTg) healthy mice (5.94±0.43 for neonates or 36.9±1.67 g for old mice) from both sexes were used in all experiments. The C57BL6 mice were from Charles River Laboratories (Wilmington, MA) and the 3xTg mice [17] were obtained as a gift from Dr. Frank Laferla (University of California at Irvine, CA, USA). Mice were kept at 21–22˚C with a 12-hour light-dark cycle with food and water *ad libitum*. This study was not pre-registered and we strictly adhered to the ARRIVE guidelines in reporting our results [18]. To minimize suffering during treatment and during the experiments, mice were kept in normal thermal and enriched environments. In addition, they were closely observed for any signs of distress, obvious pain, and discomfort.

## Isoflurane exposure

Postnatal day (PND) 7 (Neonates) and 14-16-month-old (Old) mice from both strains (3xTg-AD and NonTg) were randomly assigned to Iso (2-chloro-2-(difluoromethoxy)-1,1,1-tri-fluoro-ethane) and control groups and treated daily for 2 hours over 5 consecutive days. Briefly, Mice were placed on a padded container in plexiglass chambers, which were placed in a heated water bath to maintain a constant chamber temperature of 37±0.5˚C. This elevated chamber temperature was necessary to prevent hypothermia and to assure proper recovery from anesthesia as previously described [19]. Under those conditions, the mean rectal temperature of the animals following exposures was 37±0.5˚C. Mice were exposed to 1.5% (Neonates) or 1.1% (Old) Iso for 2h each day for five consecutive days. Repeated Iso exposure was elected because of higher probability of GAs neurotoxicity in the developing brains and associated cognitive dysfunction in both animal studies[20] and in pediatric patients[21], especially using 5 rather than 3 or 2 repeats. These concentrations represent the minimum alveolar concentration (MAC) for the age groups in our study when measured at 37˚C [19]. The calculated MAC values herein assured that the two aged groups received equipotent anesthetic throughout the treatment duration. Iso was delivered to the chambers using a vaporizer and 30% oxygen/70% nitrogen as carrier [22]. Control mice were exposed in similar conditions, but the oxygen/nitrogen gas carrier or sham was delivered without Iso. Mice breathed spontaneously without intubation and the total gas flow rate was 5 liters per minute. Iso concentration in each chamber was monitored by infrared absorbance (Ohmeda 5330, Detex-Ohmeda, USA). We did not monitor blood gases during the exposures since the daily exposure paradigm used in this study (1.5% Iso for 2h) has been previously shown to not affect arterial blood gas [23].

## S100β enzyme-linked immunosorbent assay

Blood samples from 8–10 mice were collected from the left ventricle of Iso- and sham-treated mice just prior to the transcardial perfusion described below in the brain preparation section. Cells and plasma were separated from whole blood by centrifugation for 10 min at 1500 RPM using a refrigerated centrifuge. The resulting supernatant was taken as plasma and stored at -80˚C until enzyme-linked immunosorbent assay (ELISA). S100β or calcium-binding protein beta chain levels in the plasma, an emerging marker of brain damage [24], were determined using the Sangtec 100 ELISA kit (Cat# 314701, DiaSorin, Stillwater, MN USA,) following the manufacturer's instructions.

## Brain tissue preparation

Brain samples were processed for immunohistochemical and immunoblotting studies immediately after the last of the five daily 2h exposures. Briefly, Iso- and sham-treated mice were anesthetized with sodium pentobarbital (100 mg/kg i.p.) 2 h after the end of five consecutive daily (2 hrs each) Iso exposures and transcardially perfused with cold phosphate buffered

saline (PBS, Sigma-Aldrich, USA, Cat# P5493,). The use of sodium pentobarbital rather than Iso assured that the animals would not wake up during the procedure. Following, anesthesia, the brains of completely obtunded mice were extracted and immediately dissected. The left hemisphere was frozen in liquid nitrogen and stored at −80˚C for immunoblotting, whereas the right hemisphere was fixed with 4% paraformaldehyde (Sigma-Aldrich, USA Cat# 158127,) and processed for immunohistochemistry.

## Immunohistochemistry

Mouse brains from 4 mice per group were processed for immunohistochemistry as previously described Tang, Mardini (22). Briefly, paraffin embedded coronal sections (10μm) were cut and mounted on slides, deparaffinized in xylene, and rehydrated with graded alcohols. Epitope retrieval was performed in Antigen Unmasking Solution (Vector Laboratories, USA, Cat# H-3301) by heating for 2 minutes in a decloaking chamber (Biocare Medical, USA, Cat# DC2012). Endogenous peroxidase activity was quenched with 5% hydrogen peroxide (Sigma-Aldrich, USA, Cat# H1009) and sections were then incubated in blocking solution containing 2% normal goat serum (Millipore-Sigma, USA, Cat# S26) or horse serum (Thermo Fisher Scientific, USA, Cat# 16050130) for 1hr at room temperature. Following PBS washes, sections were incubated with primary antibody 6E10 (1:400; Covance Research Products Inc, USA, Cat# SIG-39300-500, RRID:AB_662807) or AT180 (1:400, Thermo Fisher Scientific, USA, Cat# MN1040, RRID:AB_223649) overnight at 4˚C. The 1˚ antibody 6E10 was used for plaque and amyloid beta (Aβ) related peptides, whereas AT180 was used for phosphorylated tau at threonine 231 and serine 235 (pTau231/235). Sections were then incubated in either biotinylated anti-mouse or anti-rabbit IgG (1:400; Jackson Immuno Research, USA, Cat# 115-065-062 for mouse or 111-065-045 for anti-rabbit) for 1h followed by Avidin-Biotin-horseradish peroxidase Complex (1:500; Vector Laboratories, USA, Cat# PK-4000). Immunoreactivity was detected using a Diaminobenzidine kit as described by the manufacturer (Vector Laboratories, USA, Cat# SK-4103, RRID:AB_2336521) and sections were mounted on cover slides following dehydration with graded alcohols and clearing with xylene. Images were acquired with an Olympus IX70 microscope (Olympus, USA, SKU: SP-IX70FL) equipped with a Cooke Sensi-Cam camera (Applied Scientific Instrumentation, USA) using IP lab 4.0 software (Biovision Technologies, USA). At least two sections were counted from each animal and 5 animals were used in each group. Cell counts were made in 1 mm$^2$ area in the CA1 region of the hippocampus using a 20X objective field of view and given as the number of Tau or Aβ positive cells/mm$^2$.

## Immunoblotting

Hippocampal/cortical tissue was harvested from Iso- and sham-treated mice. Tissue was homogenized on ice in NP-40 (Sigma-Aldrich, USA, Cat# 9016-45-9) lysing solution supplemented with protease (Sigma-Aldrich, USA, P8340) and phosphatase (PhosSTOP; Roche, USA, Cat# 4906845001) inhibitors. Total protein was quantified by a bicinchoninic acid protein assay kit (Thermo Scientific, USA, Cat# 23225) and 60μg from each sample was separated on 6–15% sodium dodecyl sulfate-acrylamide (BioRad, USA, Cat# 1610154 for Acrylamide and 1610301 for SDS) gels. Protein was transferred onto nitrocellulose membranes (0.45μm; BioRad, USA, Cat# 1620115) at 100 V for 1h at RT. Membranes were blocked with 5% bovine serum albumin (Sigma-Aldrich, USA, Cat# A1933) and 0.1% Tween-20 in tris-buffered saline (Sigma-Aldrich, USA, Cat# T5912;TBS-T) for 1 h at room temperature and labeled with the primary polyclonal antibodies caspase 9 (Cat# 9508, RRID:AB_2068620), caspase 12 (Cat# 2202, RRID:AB_2069200), Bax (Cat# 2772, RRID:AB_329921), and Bcl-2 (Cat# 2876, RRID:

AB_2064177) from Cell Signaling Technology (All at 1:1000) and β-actin (1:2000; Santa Cruz Biotechnology, USA, Cat # sc-47778). Following washes in TBS-T, blots were incubated in secondary antibodies (anti-mouse or anti-rabbit 1:10000, Jackson Immuno Research, USA, Cat# 115-035-146 for anti-mouse or 111-035-045 for anti-rabbit). Signals were detected with enhanced chemiluminescence (GE Healthcare, USA, Cat# RPN2209) and images were acquired using the image station 4000MM Pro (Kodak, USA, Cat# 811–6634). Images were quantified using NIH image J software and normalized to β-actin.

## Spatial navigation

Spatial Cognition was assessed in 16–20 mice per group using the Morris Water Maze (MWM) tests approximately 4 weeks after the Iso exposure [25, 26]. Thus, the old mice were approximately 15 months of age and mice exposed as neonates (P7-P11) were approximately 40 days old at the start of MWM testing. This delay from last exposure to spatial cognition testing was necessary to allow the neonates to reach weanling, a period when the onset of water maze learning has been noted in mice and rats [27, 28]. Briefly, the water maze consisted of a circular swimming pool (150 cm in diameter) made of plastic. The pool was filled with water to within 15–20 cm from the top and an elevated escape platform (15-cm$^2$ Plexiglas square) was submerged 0.5–1.0 cm under the water level. To prevent the mice from seeing the submerged escape platform, the water was made opaque by the addition of titanium oxide (Sigma, USA, Cat# 481041). To facilitate acquisition and analysis, the pool was divided into 4 quadrants: east (E), west (W), north (N), and south (S). Mice were first trained to associate the escape platform with rescue using cued-response training paradigm of the MWM. For the cued trials, the pool was completely blocked from unintended cues in the room using dark curtains and the platform was fitted with a centrally mounted post (12 cm in height), which contained 2–3 flags painted with black and white horizontal stripes. Cued training was performed using four daily trials performed over 2 consecutive days. During this cued training, the escape platform was placed at a distinct quadrant (E, W, S, and N) for each of the trials and the start locations were selected in a manner that ensured each test mouse was able to locate the elevated flag-containing escape platform from drop zones near and far from it. Specifically, we used the following platform to drop zone configurations: N-SE (Southeast), E-NE (Northeast), S-SW (Southwest), W-SE, S-NE, N-NW (Northwest), W-NE, and E-SE. Each training trial allowed for a 60sec search period and each animal was allowed to remain on the platform for 15 s before the next trial. Two days after the completion of the cued-trials, we assessed the performance of the mice in reference learning to determine their ability to learn the spatial relationship between distant cues and the escape platform. Visual cues in the form of black and white geometrical shapes were placed around the pool and remained in the same position throughout the testing periods. The escape platform was then placed in the center (16 cm from the wall) of the southwest quadrant of the pool and remained there throughout the reference learning sessions. Four distinct mouse drop locations (N, SE, NW, and E) were used, but the order of the start locations was changed from day to day to prevent mice from developing fixed motor patterns. Reference testing consisted of 4 daily trials for five consecutive days, with each trial lasting 60 sec. The escape latency to platform was measured for each mouse and compared across genotypes and treatments. To measure memory recall, we assessed spatial memory using the probe test of the MWM test 1 and 24h after the completion of the last reference trial on day 5 in all mice. During this probe test, the escape platform was removed and the mice were allowed to freely swim for 30 seconds from the NE start location. The cumulative proximity to the location of the old platform for each mouse was measured and compared across genotypes and treatments. Following the probe trials, we tested working memory using

a trial-dependent learning procedure as previously described [26]. In this trial-dependent learning procedure, the platform was relocated every day and the mice were given 3 trials per day for 21 consecutive days [26]. For each day, the first trial for each mouse represented a sample (0 min) trial and two matching or test trials were given 1 and 30 min after. Mean escape latency over 21 days was recorded for each trial and given as the mean difference or time saved between the sample trials (0 min) and the 1 min (Short-term working memory) or 30 min (Long-term working memory) matching trials. Thus, a larger amount of time saved implies better working memory performance. The water in the pool was maintained at 23°C and all experiments were carried out in a dimly lit room and recorded using a video tracking system (Watermaze3, Actimetrics). To minimize bias and stress during all the MWM tests, mice were gently placed in the water facing the wall of the pool and those that failed to locate the escape platform during the allowed search period were gently guided to it. To prevent hypothermia, mice were always placed in warming cage with dry paper towel under a heat lamp upon removal from the pool for at least 5 min before returning to their home cage.

## Rotarod

Motor coordination was assessed using an accelerating rotarod test (4–40 rpm). Briefly, 16–20 mice per group were habituated to stay on the apparatus (IITC Life Sciences Rotarod Test, RRID:SCR_015698) in the stationary mode for 5 min and subsequently trained over two trials with the rotation set at a relatively slow speed (10 rpm). Following training, mice were tested in the accelerating rotarod (4–40 rpm) over 3 trials in a single day with an inter-trial interval of 30 minutes and the latencies to fall were recorded by the automatic timers and falling sensors of the apparatus.

## Extracellular field recording

**Hippocampal slice preparation.** To mitigate other anesthetic-driven confounding factors in our synaptic studies, we used Iso for acute induction of anesthesia in experimental mice (8 per group) previously exposed to sham or Iso prior to brain extraction. Following anesthesia, the brains were extracted rapidly and chilled in ice-cold dissection solution (in mM: sucrose 206, Na-Pyruvate 2, KCl 2, $NaH_2PO_4$ 1.25, $NaHCO_3$ 26, glucose 10, $MgCl_2$ 4, $MgSO_4$ 2, ascorbic acid 0.4, $CaCl_2$ 0.5; All from Sigma-Aldrich, USA). Transverse hippocampal slices (400μm) were cut with a VF300 microtome (Precisionary Instruments, USA). Intact slices were placed in a holding chamber containing aCSF (in mM: NaCl 124, KCl 2, $NaH_2PO_4$ 1.25, $NaHCO_3$ 26, glucose 10, $CaCl_2$ 2.5, $MgSO_4$ 1) at 32°C and oxygenated with 95% $O_2$/5% $CO_2$ for at least 2h before recordings. The osmolarity of all solutions was measured at 300–310 mOsm and the pH was maintained at ~7.3 under constant carbogenation.

**Extracellular field recordings.** Hippocampal CA1 field potentials were recorded on submerged slices (400μm) with a continuous flow of carbogenated (95% $O_2$/5% $CO_2$) aCSF (2 ml/min) at 34°C as previously described by Drew, Stark [29] For measurement of the CA1 field potentials, a tungsten concentric bipolar stimulating electrode (Cat# TM33CC05, 1μm tip diameter; WPI) was placed in the stratum radiatum (SR) to stimulate the Schaffer collateral (SC) pathway and a borosilicate glass recording electrode (2–3 MΩ) filled with recording aCSF was placed in the CA1 pyramidal cell layer. Electrical pulses were delivered via a pulse generator AMPI Master 8 connected to an AMPI biphasic Iso-Flex stimulator in current mode (Jerusalem, Israel). At the beginning of each recording session, an Input-output (I/O) relationship was established by increasing the stimulation intensity from 0 to 160μA in 20μA increments. For PPF, baseline, and LTP recordings, the stimulus intensity was adjusted to evoke a field excitatory postsynaptic potential (fEPSP) 2/3 of the maximum. PPF was assessed using inter-

stimulus intervals of 10, 25, 50, and 100ms. Ten successive response pairs were recorded at 0.1 Hz intervals for each interstimulus interval. Baseline (30 min) and post-tetanus (90 min) fEPSPs were recorded at 0.033 Hz and LTP was induced by a high frequency stimulation (HFS) protocol consisting of four 500ms trains of stimuli delivered at 200 Hz at test intensity and pulse duration, with 5 min between trains. All recordings were acquired at 10 kHz with pCLAMP 10 software with a Multiclamp 700B amplifier and digidata 1440A (Molecular Devices Corporation, USA). For analysis, fEPSPs for I/O relationship, baseline, and LTP recordings were quantified by measuring the slope of the initial rising phase of the response in ClampFit 10 (Molecular Devices Corporation, Sunnyvale, CA, USA). For LTP recordings, fEPSP slopes recorded after the HFS protocol were averaged and expressed as a percentage of the average slope from 30 min baseline recordings. For PPF, fEPSP amplitudes were expressed as the ratio of the 2nd fEPSP over the first fEPSP.

## Data analysis and statistics

Data analysis was done blindly without knowledge of treatment condition of each animal using GraphPad Prism (v5.0 for Windows, GraphPad Software, USA) and given as mean ± SEM. The number of animals used in all the experiments is reported in the figure legends and was based on numbers used for similar experiments in the literature as well as power analysis. We did not perform any test for outliers and there were no sample size differences between the beginning and end of our experiments. All immunohistochemistry, Western blotting, and S100β ELISA, probe trials, and working memory datasets were analyzed using one-way analysis of variance (ANOVA) tests with Tukey's multiple comparison post hoc tests. A two-way (Repeated measures) ANOVA with Bonferroni post hoc tests was used to compare reference learning data. Finally, all the electrophysiology data sets were analyzed using two-way ANOVA followed with Bonferroni post hoc tests. The significance level for all of our analyses was set at 95% ($P < 0.05$).

## Results

### Effects of repeated isoflurane exposures on amyloid load and tauopathy

Accumulating evidence in the literature suggests that exposures of various AD mouse models to anesthetics can lead to immediate cellular pathologies followed by a diverse array of cognitive impairments later in life [9]. Here, we tested the hypothesis that repeated exposures of the 3xTg mice to Iso will lead to immediate cellular pathologies and diverse cognitive changes that are associated with different aspects of synaptic functions. To that end, we exposed neonatal and old mice from both 3xTg and NonTg genotypes to Iso and tested our hypothesis in a series of experiments as illustrated in the experimental timeline (Fig 1A). Mice began treatments at P7 or 14 months of age and the start of treatment is indicated as day 0 rather than the age of the mice (Fig 1A). We first assessed neuropathology in the form of amyloid load. Quantitative analysis of amyloid load showed that intracellular amyloid deposits labeled with the anti-Aβ 6E10 antibody were noted in neurons throughout the CA1 region of the hippocampus of sham- and Iso-treated old 3xTg mice (Fig 1A–1D). As expected, very few 6E10 positive neurons were seen in the CA1 of sham-treated neonates and old NonTg mice (Fig 1A and 1B). Interestingly, repeated Iso exposures significantly increased the total number of 6E10 positive cells in the CA1 region of the hippocampus in 3xTg neonates (Fig 1D; $p < 0.0004$), but not in old 3xTg mice (Fig 1E; $p > 0.05$). Surprisingly, there was no significant difference in the number of 6E10 positive cells in the hippocampus of sham-treated old 3xTg-AD mice compared to age-matched and sham-treated NonTg mice, but there was a strong trend toward more 6E10 positive cells in the old 3xTg-AD mice (Fig 1E; $p > 0.05$). These results suggest that neonatal

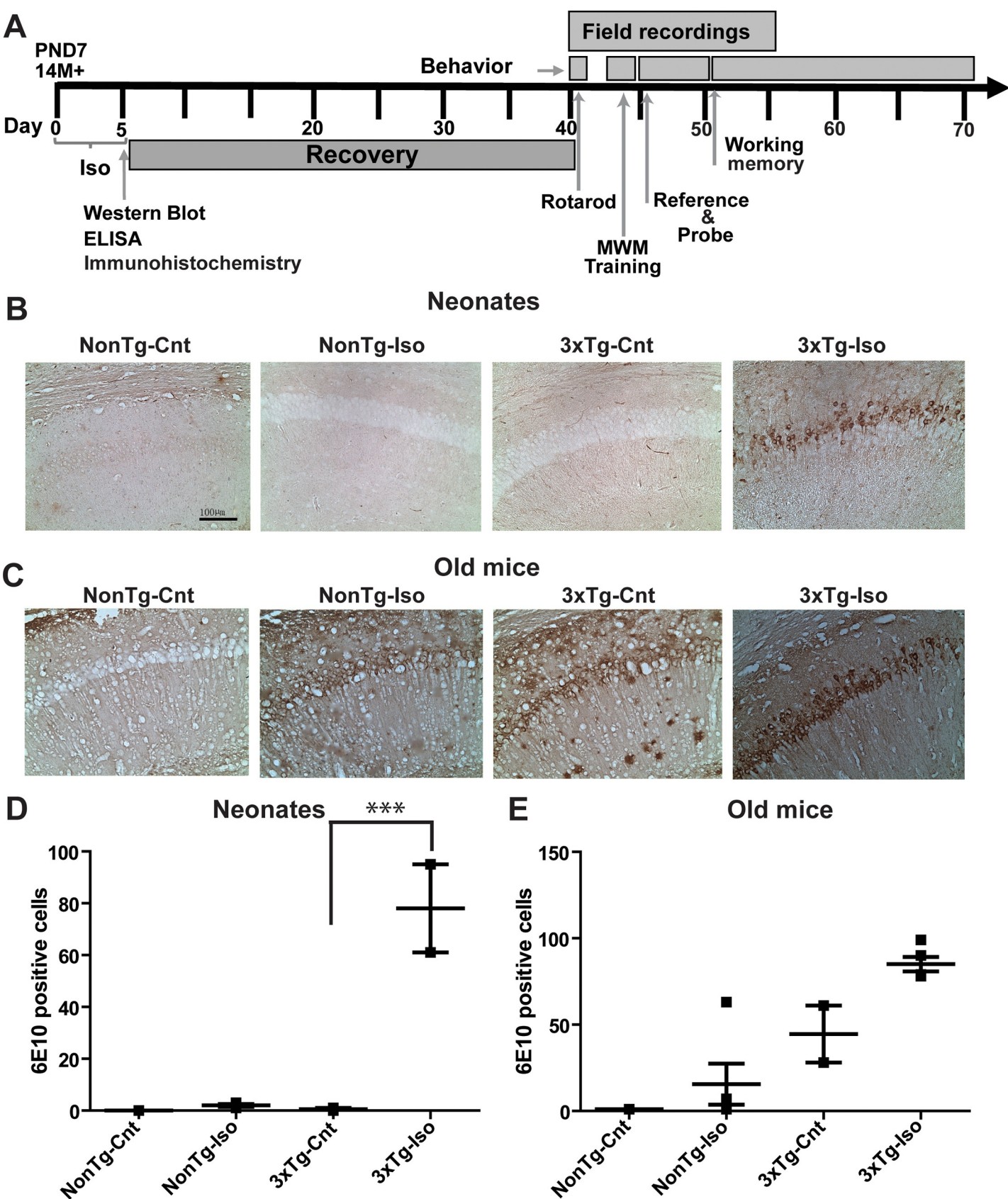

**Fig 1. Isoflurane increased amyloid burden in the hippocampus of neonatal and old 3xTg-AD mice.** A) Timeline of the experimental plan, with arrows indicating the start of each experimental paradigm. Representative micrographs of 6E10 labeling in the CA1 region of the hippocampus of neonatal (B) and old (C) NonTg or 3xTg-AD mice following repeated sham or Iso exposures. Quantitative analysis of 6E10 positive cells in neonates (D) and old mice (E) mice. Statistical significance indicated with asterisks: ***p<0.001. N = 4 animals per group. Scale bar is 100µm.

3xTg mice may be more susceptible to Iso-mediated increase in amyloid load than the old 3xTg mice.

To further assess the impact of Iso on immediate cellular pathology, we measured tauopathy using the Tau specific antibody AT180. As expected, we noted very few AT180 positive neurons in hippocampal CA1 area of neonatal and old NonTg mice in both sham and Iso-treated groups (Fig 2A–2D; $p > 0.05$). This low level of AT180 and non-significant immunoreactivity was similarly noted in neonatal 3xTg mice exposed to sham or Iso (Fig 2A–2C; $p > 0.05$). In contrast, both sham- and Iso-treated old 3xTg mice showed prominent AT180 labeling in the CA1 hippocampal region (Fig 2B). However, Iso-exposed old 3xTg mice showed significantly more AT180 positive cells compared to sham controls (Fig 2D; $p < 0.001$). Taken together, these observations suggest that repeated Iso exposures may exacerbate Tau neuropathology in 3xTg mice.

As a further measure of cellular pathology, we investigated the impact of repeated Iso exposures on the plasma levels of S100β. Sham-exposed neonatal 3xTg and NonTg mice displayed the same levels of S100β (Fig 2E and 2F; $p > 0.05$). Repeated exposures of these neonatal mice to Iso resulted in an increase in S100β in NonTg (Fig 2E; $p < 0.0004$), but the levels of S100β in 3xTg mice were not affected (Fig 2F; $p > 0.05$). Interestingly, repeated Iso exposures did not alter the levels of S100β in old NonTg (2F; $p > 0.05$) and 3xTg (Fig 2F; $p > 0.05$) compared to their respective age-matched sham controls. These results suggest repeated Iso exposures promoted S100β release only in the bloodstream of neonatal NonTg mice.

## Repeated isoflurane exposures induced apoptosis in mouse brains in an age-dependent manner

Previous studies have suggested that single or repeated Iso exposures may induce apoptosis [30, 31]. To further investigate the impact of repeated Iso exposures on neuropathology, we measured the expression levels of the apoptotic markers caspase-9 and caspase-12 as well as the ratio of Bax/B-cell lymphoma 2 (Bcl-2) using quantitative Western blot analysis. Repeated Iso exposures led to a near three-fold increase in caspase 9 intensity in 3xTg-AD neonates (Fig 3A and 3C; p < 0.02), whereas neonatal NonTg mice only trended toward an increase in caspase-9 (Fig 3A and 3C; $p > 0.05$). In sharp contrast to caspase-9 expression, quantitative analysis of the apoptotic/ER stress marker caspase-12 showed a near two folds increase in intensity in neonatal NonTg (Fig 3A and 3D; $p < 0.0007$), whereas neonatal 3xTg mice only trended toward an increase in caspase-12 following repeated Iso exposures (Fig 3A and 3D; $p > 0.05$). To further interrogate the impact of repeated Iso exposures on apoptosis, we measured the ratio of Bax/Bcl-2, two Bcl-2 family proteins that play key roles in promoting or inhibiting the apoptotic pathways triggered by mitochondrial stress [32, 33]. While the ratio of Bax/Bcl-2 in neonatal NonTg mice was comparable in Iso- and sham-treated animals, (Fig 3E; $p > 0.05$), we noted a significant increase in Bax/Bcl-2 ratio in 3xTg neonates following repeated Iso exposures (Fig 3E $p < 0.0003$). In contrast to the neonates, repeated Iso exposures did not alter the expression of caspase-9 (Fig 3G; $p > 0.05$), caspase-12 (Fig 3H; $p > 0.05$), nor the Bax/Bcl-2 ratio (Fig 3I; $p > 0.05$) in the old mice in either genotype. This lack of Iso-mediated effects on apoptosis in the old mice is consistent with previous studies [14, 34]. Taken together, these results suggest apoptosis induction in neonatal 3xTg mice showed a preference for the

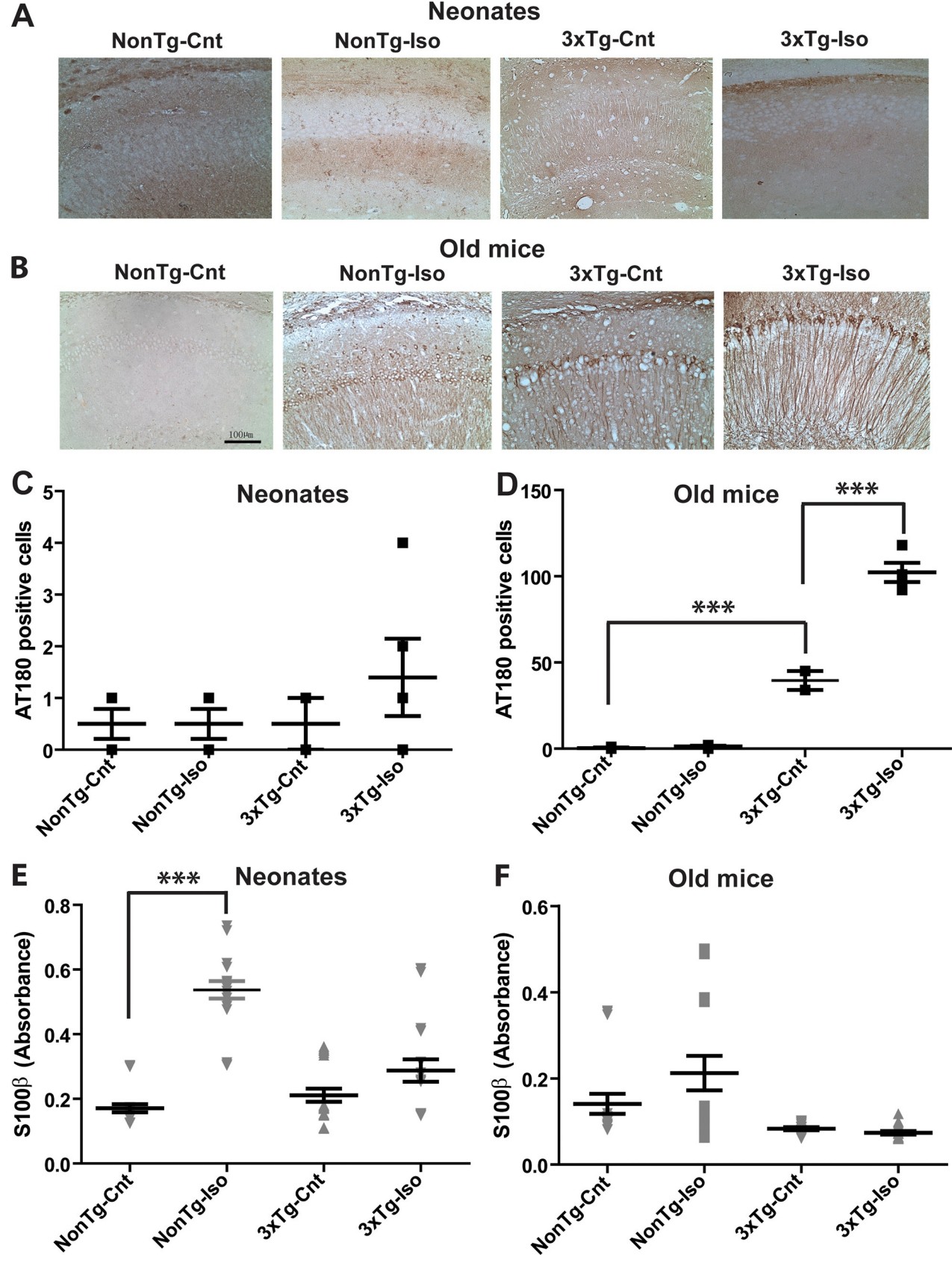

**Fig 2. Isoflurane selectively worsened Tau pathology in old 3xTg-AD mice and S100β release in neonatal NonTg mice.** Representative micrographs of AT180 positive cells in the CA1 region of the hippocampus of neonatal (A) and old (B) NonTg or 3xTg-AD mice following repeated sham or Iso exposures. Quantitative analysis of AT180 positive cells in the CA1 hippocampus of neonates (C) and old mice (D). Plasma level of S100β in neonates (E) and old mice (F). Statistical significance indicated with asterisks: ***p<0.001. N = 4 animals per group and counts are given as cells/mm². Scale bar is 100μm.

activation of the caspase-9 dependent mitochondrial pathway and the neonatal NonTg mice exhibited a preference for the caspase-12 mediated ER-stress pathway.

## Effects of repeated isoflurane exposures on spatial learning and memory

A number of experimental reports have confirmed that the effects of prolonged anesthetic exposures on learning and memory in neonatal and old rodents, be it improvement or deterioration of cognitive functions, can last for weeks [9]. Thus, we first asked whether repeated Iso exposures led to long-term spatial cognitive deficits in mice exposed as neonates and old in both NonTg and 3xTg groups 30 days after the last exposure. Mice exposed to Iso or sham as neonates and tested 30 days later are hereafter referred to as Iso- or sham-treated neonates. Sham-exposed neonatal NonTg mice performed significantly faster than sham-exposed 3xTg mice during the first two days of testing, suggesting that the triple transgenes may have some degree of early impaired baseline learning ability (Fig 4A; *, **$P < 0.05, 0.01$, respectively). While reference learning in neonatal NonTg mice was mostly unaffected by repeated Iso exposures, neonatal 3xTg mice exposed to Iso in the same manner were able to locate the hidden platform with significantly longer latencies on days 3 to 5 compared to sham controls (Fig 4A; ##, ###$P < 0.01, 0.0001$, respectively). In contrast to neonates, sham- and Iso-treated old mice from both genotypes did not differ in spatial learning (Fig 4B; $p > 0.05$). These observations suggest that Iso selectively disrupted learning only in neonatal 3xTg mice.

Immediately after reference learning, we tested short- and long-term memory retention using the probe trials paradigm of the MWM test 1 and 24h after completion of the last reference trial in all the mice. Given that the spatial distribution of an animal location relative to the escape platform in the MWM was found to be more sensitive to age-related impairment in an assessment of young and old rats than other measures [35], we measured the cumulative distance of each animal from the platform as a measure of performance. Accordingly, a shorter cumulative proximity to the previous location of the escape platform is taken as a measure of improved performance. In the short-term probe trials conducted 1h after the last acquisition sessions of reference learning, the cumulative proximity to the platform in Iso-treated neonatal NonTg and 3xTg mice did not differ from sham controls (Fig 4C, $p > 0.05$), indicating that short-term memory retention was preserved in neonatal mice from both genotypes following repeated Iso exposures. Long-term probe trials demonstrated similar preservation of memory retention in neonatal NonTg and 3xTg mice based on the similar cumulative proximity to the previous location of the escape platform following repeated Iso exposures (Fig 4E; $p > 0.05$). As noted in the neonatal mice, the cumulative proximity measurements during the short-term probe trials did not differ between Iso- and sham-treated old NonTg and 3xTg mice in the 24h probe trials (Fig 4D; $p > 0.05$). This lack of Iso-mediated effects on short-term memory retention was also noted in the long-term probe tests in these old mice (Fig 4F; $p > 0.05$). However, comparison of sham-treated old NonTg and 3xTg mice showed that 3xTg mice were on average significantly closer to the escape platform during long-term memory testing mice (Fig 4F; **$P < 0.01$). These results suggest that the presence of the triple transgenes improved long-term memory retention and that Iso had no measurable impact on memory retention in mice from both age groups and genotypes.

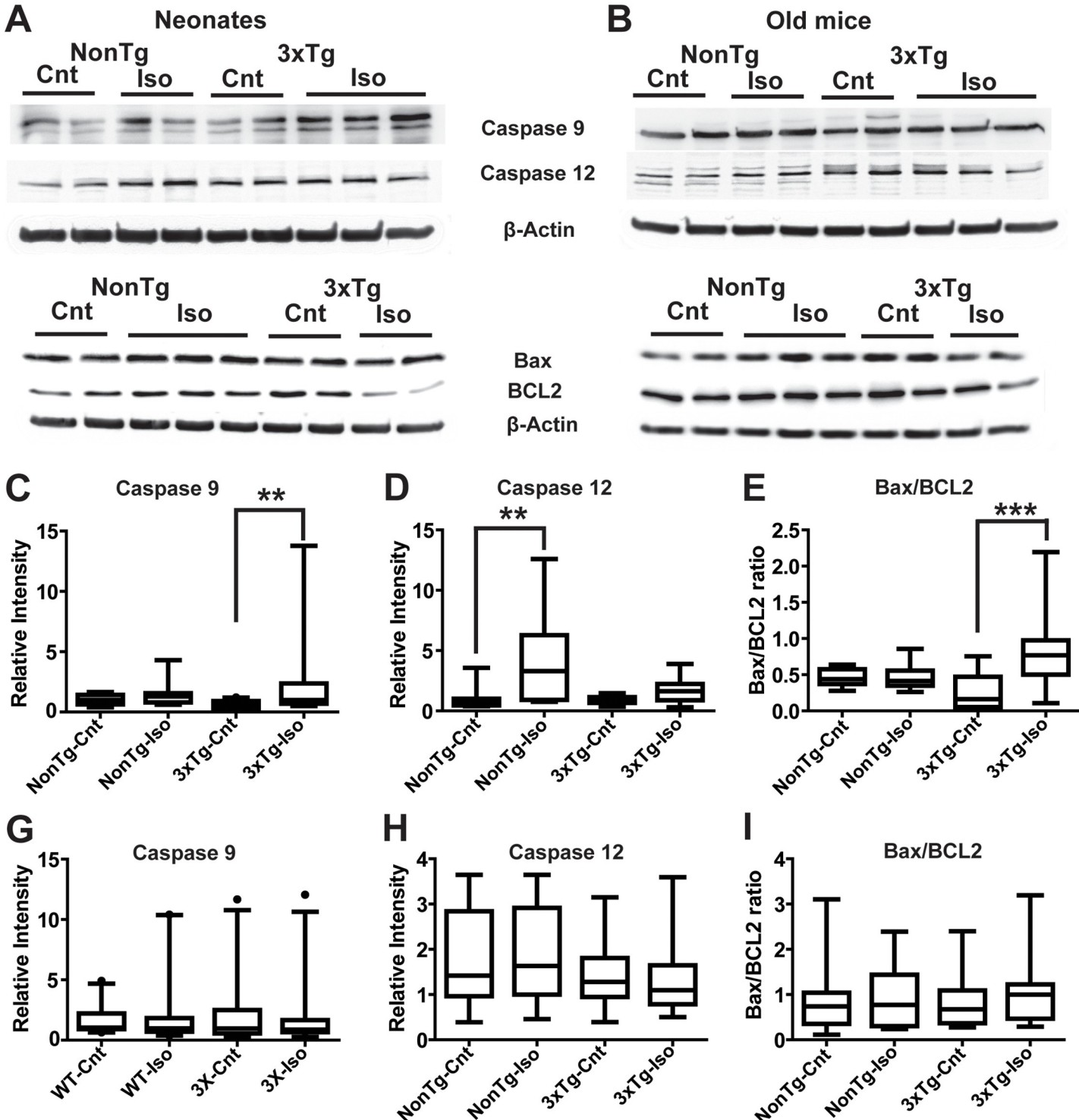

**Fig 3. Isoflurane selectively increased the expression level of apoptotic markers in neonates.** Representative Western blotting images of caspase 9, caspase 12, Bax, and BCL-2 from neonates (A) and old mice (B). Densitometry analysis of the caspase 9 (C), caspase 12 (D), Bax/BCL-2 (E) bands relative to their respective β-actin loading control band in neonates. Densitometry analysis of the caspase 9 (G), caspase 12 (H), Bax/BCL-2 (I) bands relative to their respective β-actin loading control band in the old mice. Statistical significance indicated with asterisks: **p<0.01 and ***p<0.001. N ≥8 animals per group and counts are given as cells/mm$^2$. Scale bar is 100μm.

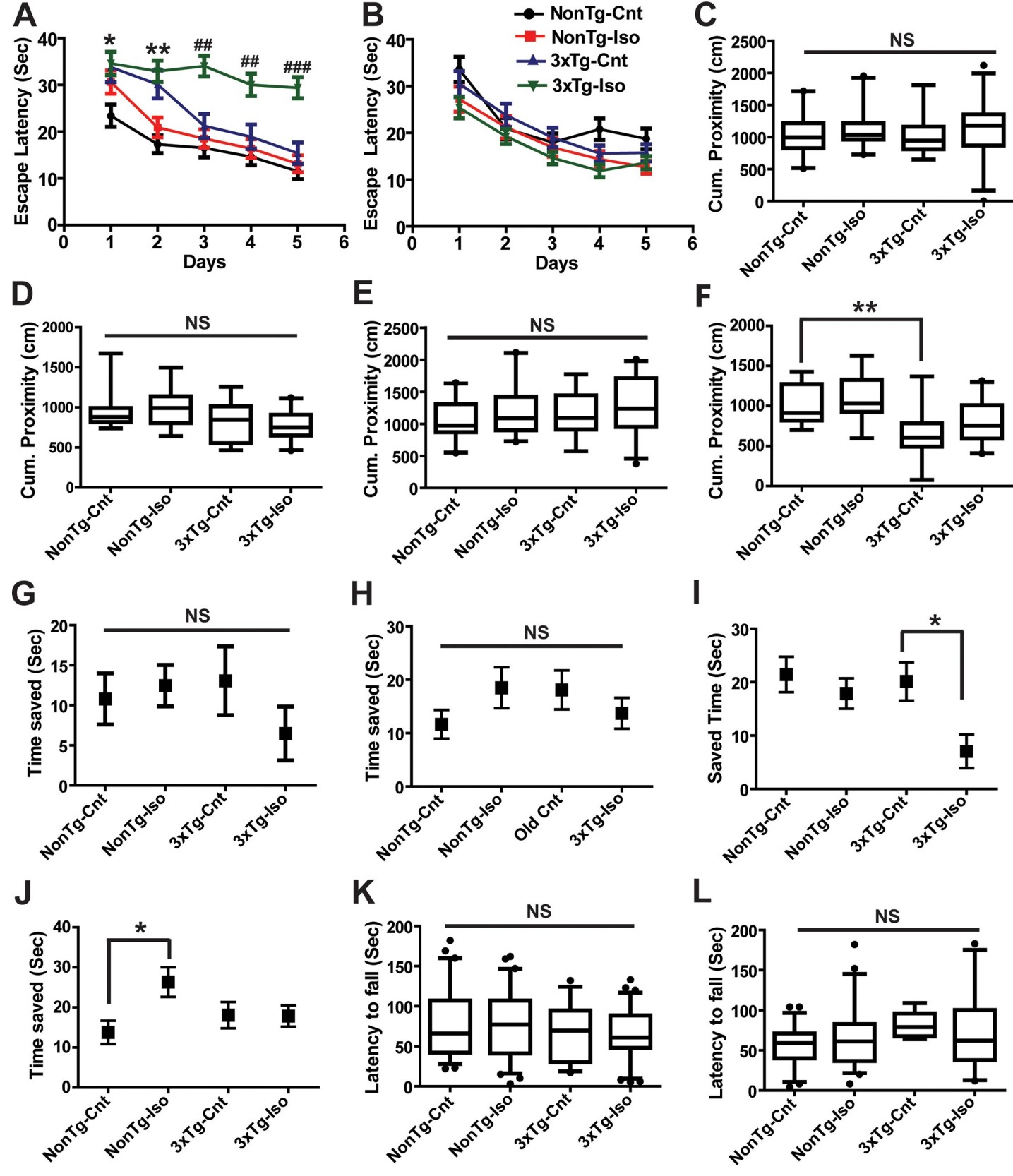

**Fig 4. Isoflurane altered distinct spatial learning paradigms in young and old 3xTg-AD and NonTg mice.** Latency to find the submerged platform in the reference learning of the MWM test during 5 consecutive days after repeated exposures of neonates (A) and old mice (B). Quantitative analysis of short-term (C) and long term (E) probe tests in neonates. Analysis of short-term (D) and long term (F) probe tests old mice. Time saved in short-term trial-dependent working memory paradigm in neonates (G) and old mice (H). Quantitative analysis of time saved during long-term trial-dependent working memory paradigm in neonates (I) and old mice (J). Motor function measured by the accelerated Rotarod paradigm was not altered by repeated Iso exposures in both neonatal (K) and old (L) mice. Statistical significance indicated with the following symbols: *p<0.05; **, ## p<0.01, ***, ###p<0.001. Statistical analyses denoted by # represent comparisons between sham and Iso treated 3xTg-AD mice and those denoted by * compared sham treated 3xTg and NonTg mice. N≥14 animals per group.

As a further measure of spatial cognition, we assessed working memory using a trial-dependent learning test as described in the methods section [26]. In this test, working memory performance is measured as the difference in the latency to the escape platform between a baseline trial and two subsequent test trials given 1(Short-term) and 30 min (Long-term) apart. This difference in latency was taken as time saved, with a larger time saved as an indicator of superior working memory. Sham-treated neonatal NonTg and 3xTg mice saved relatively equal amount time during the short-term working memory testing (Fig 4G; $p > 0.05$). This lack of effects in short-term working memory performance was similarly noted in both genotypes following repeated Iso exposures (Fig 4G; $p > 0.05$). As noted in the short-term paradigm, sham-treated neonatal NonTg and 3xTg mice saved relatively equal amount of time during long-term working memory (Fig 4I; $p > 0.05$), but Iso impaired this long-term working memory in the neonatal 3xTg mice as measured by the reduction in time saved. (Fig 4I; $p > $ *$P < 0.05$). Similar to the neonates, sham-treated old NonTg and 3xTg mice did not differ in the amount of saved time during short-term working memory testing (Fig 4H; $p > 0.05$). Additionally, repeated Iso exposures had no discernible effects on time saved during short-term trials (Fig 4H; $p > 0.05$). As noted during short-term working memory, old NonTg and 3xTg displayed similar level of performance in the long-term working memory as determined by the relatively equal amount of time saved (Fig 4J; $p > 0.05$). Although repeated Iso exposures did not significantly influence the performance of old 3xTg (Fig 4J; $p > 0.05$), old NonTg mice treated with Iso saved significantly more time during long-term working memory assessment (Fig 4J; *$P < 0.05$), indicating an enhancement in long-term working memory. The noted Iso effects on spatial learning and memory could not be attributed to motor function impairments as no significant differences were found in the accelerated rotarod test between NonTg and 3xTg mice in both neonates (Fig 4K; $p > 0.05$) and old mice (Fig 4L; $p > 0.05$) following repeated Iso exposures. Taken together, these results suggest that repeated Iso exposures impaired long-term working memory in neonatal 3xTg mice while improving those measures in old NonTg mice.

## Effects of repeated isoflurane exposures on input-output current relationship

Neonates and old mice from both genotypes increased their slope fEPSP amplitudes with current intensities (Fig 5). The fEPSP slope amplitudes at high current intensities were significantly depressed in sham-treated neonatal 3xTg mice compared to sham-treated neonatal NonTg mice, suggesting that hippocampal networks of pre-symptomatic AD mice were hypo-excitable (Fig 5A and 5C; ***p<0.001). There were no significant differences in slope amplitudes between sham- and iso-treated neonatal 3xTg mice (Fig 5C; $p > 0.05$), but Iso significantly depressed the fEPSP slope amplitudes in neonatal NonTg mice compared to age-matched sham-treated NonTg mice (Fig 5A and 5C; ##, ###p< 0.01 and 0.001, respectively). In contrast to the hypo-excitability in neonatal 3xTg mice, the slope amplitudes in the old group were significantly enhanced in sham-treated 3xTg compared to NonTg mice receiving the same treatment, suggesting that hippocampal networks of post-symptomatic AD mice were

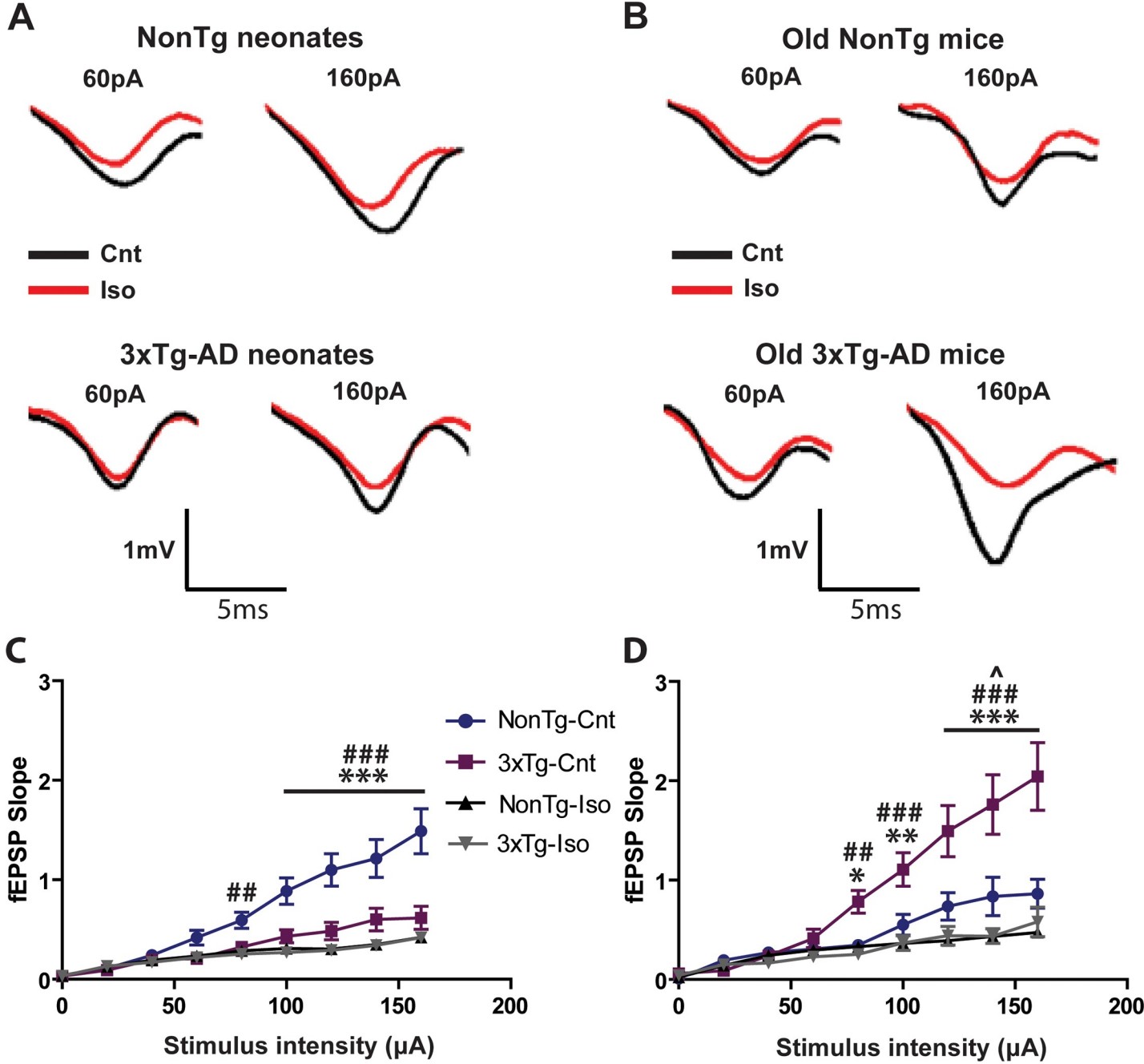

**Fig 5. Repeated *in vivo* isoflurane exposures differentially altered basal synaptic transmission in NonTg and 3xTg-AD mice.** (A) Representative traces of CA1 field EPSPs recorded in response to SC stimulations on acute slices from sham- or Iso-treated neonatal NonTg (Top traces) and 3xTg-AD (Bottom traces) mice. (B) Representative traces of CA1 field EPSPs recorded as in panel A on acute slices from sham- or Iso-treated old NonTg (Top traces) and 3xTg-AD (Bottom traces) mice. Quantitative analysis of Input-output generated by plotting mean fEPSP slope to increasing current intensities in neonatal (C) and old mice (D). Statistical significance indicated with the following symbols: *,^p<0.05; **, ## p<0.01, ***, ###p<0.001. Statistical analyses denoted by # represent comparisons between sham- and Iso-treated NonTg mice and those denoted by * compared sham-treated 3xTg and NonTg mice. Statistical analyses denoted by ^ represent comparisons between sham- and Iso-treated old NonTg mice. N≥8 animals and ≥16 slices per group.

hyper-excitable (Fig 5B and 5D; *, **,***p<, 0.05, 0.01, and 0.001 respectively). Repeated Iso exposures significantly depressed the slope amplitudes at larger intensities in both old NonTg (Fig 5B and 5D; ^p<0.05) and 3xTg-AD (Fig 5B and 5D; ##, ###p<0.001) compared to their

respective age-matched sham controls, but the extent of this depression in basal transmission was more severe in the old 3xTg mice (Fig 5D; ^p<0.05 Vs ###p<0.001). These observations suggest CA1 neural networks in 3xTg mice become more excitable in older animals and that repeated Iso exposures more profoundly impaired basal synaptic transmission in post-symptomatic 3xTg mice.

## Short-term synaptic plasticity

Paired-pulse ratio (PPR) of slope amplitude response at a paired-pulse interval (PPI) of 150ms was significantly less facilitated in sham-treated neonatal 3xTg mice compared to sham-treated neonatal NonTg mice (Fig 6A and 6C; **p<0.01). Given the inverse relationship between PPF and release probability [36], these results suggest that CA1 microcircuits in neonatal 3xTg consist of synapses with high release probability of neurotransmitters. There was a significant reduction in the fEPSP slope amplitude facilitation at 100 and 50 ms PPI by repeated exposures of neonatal NonTg mice to Iso (Fig 6A and 6C; ##p<0.01). However, we did not observe any significant differences in PPF between sham- and Iso-treated neonatal 3xTg mice at any PPI (Fig 6C; *p* > 0.05). In stark contrast to the neonatal 3xTg mice, sham-treated old 3xTg mice showed significantly more facilitated paired pulse responses at PPI of 50, 100, and 150 ms compared to sham-treated old NonTg mice (Fig 6B and 6D; ***p<0.001 for 50 and 100 PPI; **p<0.01 for 150 PPI), suggesting CA1 synapses in 3xTg mice switched from high release probability in neonates to low release probability synapses in old mice, whereas the NonTg mice switched from low release probability in neonates to high release probability synapses in old mice. There were no significant differences between sham and Iso-treated old NonTg mice (Fig 6D; *p* > 0.05), but repeated exposures of 3xTg mice to Iso significantly depressed the PPF responses at PPI of 50, 25, and 10 ms when compared to sham-treated old 3xTg mice (Fig 6B and 6D; ###p<0.001 for 50 and 100 PPI; ##p<0.01 for 150 PPI). These results suggest that the triple transgenes altered the extent of short term plasticity in an age-dependent manner by promoting the expression of low facilitating CA1 synapses in neonates and high facilitating ones in older mice. In spite of the noted differences in the weight of synaptic facilitation, the results are consistent with a role for anesthetics in dampening network excitability in both neonatal NonTg and old 3xTg-AD mice.

## Long-term potentiation

As shown in Fig 7, HFS stimulation of SC in CA1 SR reliably induced comparable LTP in NonTg and 3xTg-AD mice previously exposed to air or Iso as neonates (Fig 7A, 7C and 7E; 41.3±1.4% and 37.1±3.7% above a baseline set at 100% for NonTg and 3xTg mice, respectively). A month after repeated Iso exposures, stable LTP could not be induced in neonatal nonTg (Fig 7A, 7C and 7E; 5.7±2.05% above baseline), but was preserved in 3xTg mice (Fig 7A, 7C and 7E; 35.9±2.3% above baseline), with the slope amplitude significantly depressed compared to aged-matched 3xTg and NonTg controls (Fig 7C and 7E; ***p<0.0001). However, baseline fEPSP slope in the old group was rather unstable and HFS stimulation initially triggered a depression in the slope amplitude and slowly rising LTP of the slope amplitude in untreated 3xTg mice compared to untreated NonTg mice (Fig 7B, 7D, and 7F; ***p<0.0001). LTP could not be induced by HFS in old NonTg mice following Iso treatment (Fig 7B, 7D and 7F; 4.8±1.45% above baseline), with the slope amplitude significantly depressed in Iso-treated NonTg mice compared to shams (Fig 7D and 7F; ***p<0.0001). By contrast, robust LTP could be induced in aged 3xTg mice following such treatment (Fig 7B, 7D and 7F; 73.6±0.92% above baseline). In fact, the early phase of the potentiation in Iso treated old 3xTg mice was significantly enhanced compared to sham-treated 3xTg controls (Fig 7D and 7F; ***p<0.0001).

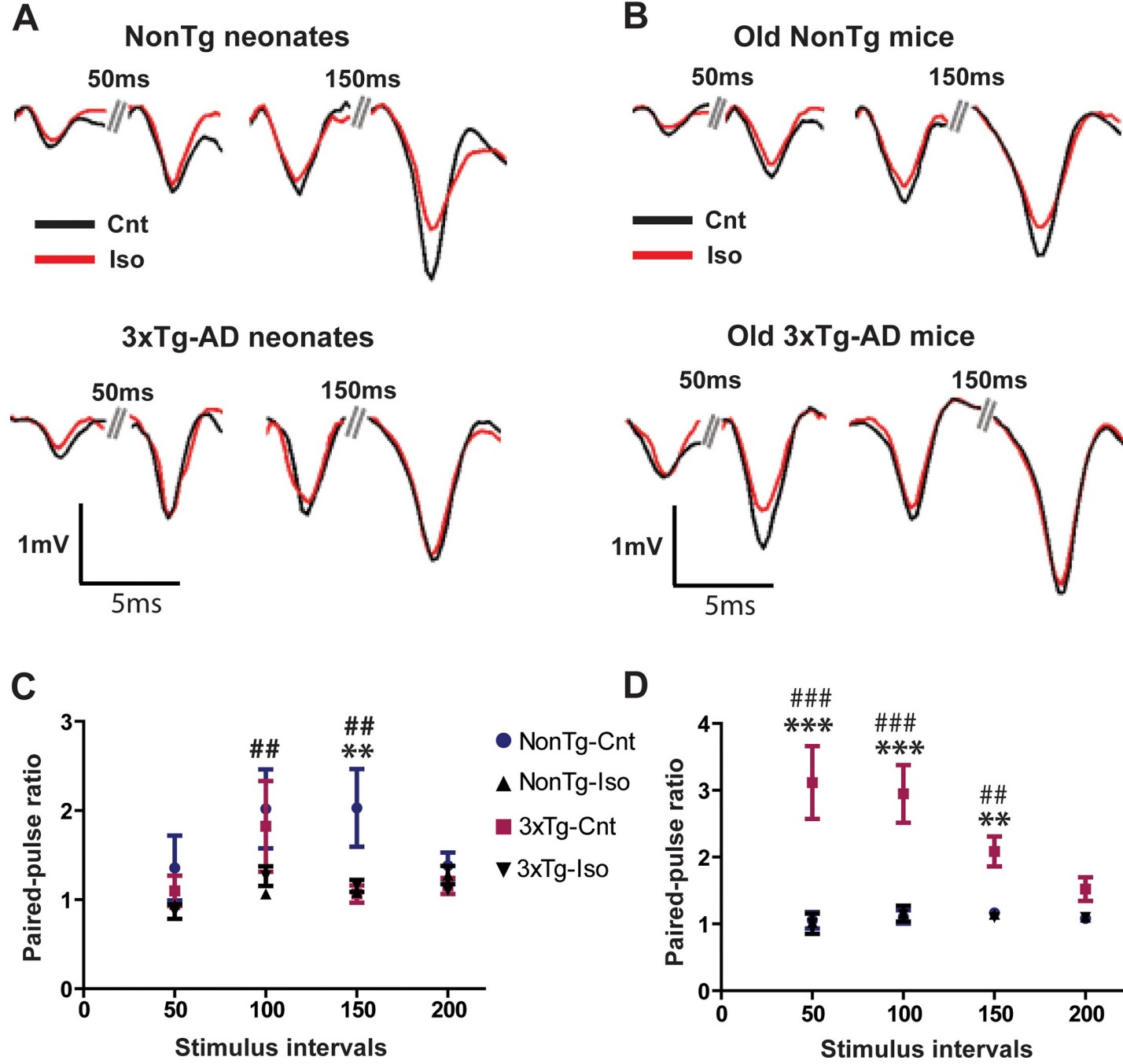

**Fig 6. Paired-pulse ratio is differentially altered Iso exposures.** (A) Representative traces of paired CA1 field EPSPs recorded at various inter-stimulus intervals (ISI) on acute slices from sham- and Iso-treated neonatal NonTg (Top traces) and 3xTg-AD (Bottom traces) mice. (B) Representative traces of paired-pulse CA1 field EPSPs recorded as in panel A on acute slices from old NonTg (Top traces) and 3xTg-AD (Bottom traces) mice. Quantitative analysis of fEPSP slope PPR (Pulse 2/Pulse 1) in neonatal (C) and old mice (D). Statistical significance indicated with the following symbols: **, ## $p<0.01$, ***, ### $p<0.001$. Statistical analyses denoted by # represent comparisons between sham- and Iso-treated NonTg mice and those denoted by * compared sham-treated 3xTg and NonTg mice. N≥8 animals and ≥16 slices per group.

Taken together, these results suggest that isoflurane normalized synaptic plasticity defects in old 3xTg mice, while exacerbating those defects in early pre-symptomatic mice.

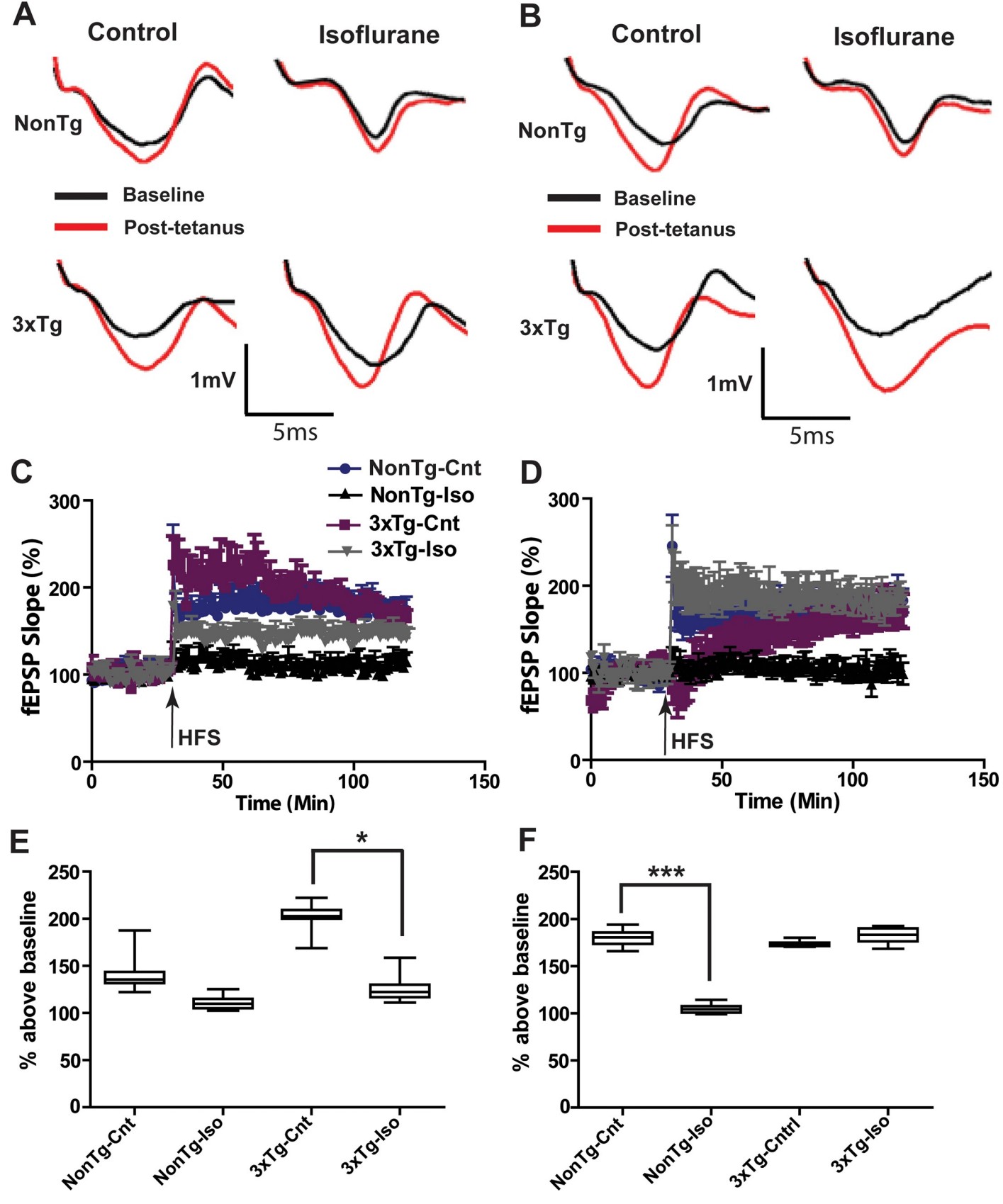

**Fig 7. Repeated *in vivo* isoflurane exposures differentially altered LTP in NonTg and 3xTg-AD mice.** (A) Representative traces of CA1 field EPSPs recorded in response to SC stimulations before and after LTP induction on acute slices from sham- and Iso-treated neonatal NonTg (Top traces) and 3xTg-AD (Bottom traces) mice. (B) Representative traces of CA1 field EPSPs recorded as in panel A on acute slices from old NonTg (Top traces) and 3xTg-AD (Bottom traces) mice before and after LTP induction. (C) Normalized fEPSP slope before and after LTP induction (200Hz, 500ms) in sham- and iso-treated neonatal (C) and old (D) mice. (E) Quantitative analysis of the first 10 min of fEPSP slope following LTP induction on acute slices from sham and Iso-treated neonatal (E) and old (F) mice. Statistical significance indicated with the following symbols: *p<0.05; **p<0.01, ***p<0.001. N≥8 animals and ≥16 slices per group.

## Discussion

Given that early pathological alterations at the synapse may precede the gross anatomical changes in AD, we investigated the impact of repeated Iso exposures on synaptic function in 3xTg mice. We provide evidence to support the hypothesis that anesthetic exposures can increase neuropathology and impair synaptic functions in AD mice in an age-dependent manner. Our results showed that repeated exposures of neonatal and old 3xTg to Iso distinctly induced neurodegeneration. Furthermore, we demonstrated that these repeated Iso exposures can impair as well as improve some spatial cognitive tests. Finally, we show that Iso reduced network excitability and impinged on synaptic plasticity in an age-dependent manner. STP measurements suggest that the synaptic effects were likely due to dysregulation in the synaptic mechanisms mediating neurotransmitter release. These results suggest that repeated Iso exposures during the pre-symptomatic and post-symptomatic phases differentially alter the progression of neuropathology and synaptic dysfunctions in AD.

### Altered neuropathology

Repeated Iso exposures resulted in a significant increase the number of Aβ and Tau positive neurons as previously reported for Tg2576 mice [20]. These results in pre-symptomatic neonatal 3xTg mice is consistent with previous report for pre-symptomatic 2, 4, and 6-month-old 3xTg mice [22]. Unlike that study, we noted a parallel enhancement in amyloidopathy with tauopathy, albeit in a much younger cohort of mice (P7 neonates). In contrast to Tang et al. Tang, Mardini (22), we noted that repeated Iso exposures of post-symptomatic adult mice at 14 months of age increased neuropathology in 3xTg. The differences in the trajectory of Iso-mediated neuropathology between our study and Tang et al.Tang, Mardini (22) are likely due to differences in exposure paradigms and age. Interestingly, we did not detect amyloid plaques as previously noted in the 3xTg mouse line [37, 38]. These discrepant observations are likely due to our antigen retrieval method, which lacked the necessary formic acid for optimal plaques staining. The availability of Aβ epitopes to conformational antibodies depends on their particular aggregation state [39–42]. This is particularly the case for 6E10 epitopes, which are largely known to hide in some types of Aβ aggregates *in vitro* [43] and become more available with increasing formic acid concentrations as evidenced by corresponding increase in total Aβ immunoreactivity [37, 44]. We also noted a lack of significant differences in intra-neuronal Aβ immunoreactivity in the old 3xTg mice compared to NonTg mice. This observation could be explained by age-dependent changes in Aβ immunoreactivity. Notably, intracellular Aβ immunoreactivity against the more recent Aβ antibody M78 in 12 month old 3XTg disappeared in 14 month old mice and became exclusively extracellular plaques [37], suggesting dynamic changes to conformational state during aging. These observations are very much in line with previous findings showing a decrease in intraneuronal Aβ immunoreactivity with increasing cognitive dysfunction and increasing amyloid plaque deposition [45].

In addition, we noted a few labeled 6E10 cells in old NonTg mice. This observation is rather unexpected given AD transgenic models are designed to overexpress mutated human APP, allowing for the discrimination of the amyloidogenic hAPP from the non amyloidogenic

rodent APP. Indeed, the 6E10 antibody has been described as capable of discriminating between these two forms of APP in immunohistochemistry [46], but this was validated in reducing Western blot conditions rather than in non-reducing immunohistochemistry conditions [47], suggesting that the 6E10 antibody may not provide the necessary discrimination between murine and hAPP for accurate assessment of Aβ neuropathology in immunohistochemical studies. Future immunohistochemical studies should consider the novel 1D1 and 7H6 antibodies as both appear to only label APP-transgenic cells in non-reducing conditions [47]. Nonetheless, the noted Aβ and Tau observations in combination with the selective increased expression of apoptotic markers in neonates are consistent with Iso-mediated effects on neuropathology as previously noted in previous reports [11, 48].

## Spatial memory impairments

Various studies have noted unaltered cognitive performance following anesthetic exposures [49, 50], while others have not reported impairments or enhancements in similar cognitive tasks [12, 51–55]. Here, we found that repeated Iso exposures of pre-symptomatic 3xTg mice and NonTg beginning at PND 7 resulted in impaired reference learning and working memory a month after the last exposure. Although repeated exposures of old 3xTg and NonTg mice resulted in very little impairments of spatial cognitive measures, we made several unexpected observations.

First we noted that the neonatal and old mice displayed similar spatial learning performance. This lack of aging impact on performance is quite remarkable given that it can profoundly impact cognitive abilities [56–58]. Despite the fact that age-related decline is prevalent enough to be considered a normal part of aging, our results suggest that older animals are capable of retaining strong spatial cognitive abilities. Indeed, this preservation of cognitive ability has been noted in older individuals and laboratory animals [59–63].

Secondly, we noted the appearance of spatial cognitive impairments in approximately 1month old sham-exposed 3xTg mice. The timeline of the appearance of these cognitive deficits is much earlier than the 3–6 months of age timeline for the emergence of Tau and amyloid pathology noted in the 3xTg mice [17, 64]. Because we only assessed Tau and amyloid pathology at around P12 in neonates, we cannot rule out the possibility that these pathologies were present at the time of our cognitive testing. Indeed, 6E10 immunoreactivity has been noted in P21 in 3xTg mice [65] in contrast to earlier reports [17, 64]. It is not clear what the basis of these discrepant results might be, but conformational diversity of Aβ may be a contributing factor

Finally, our old 3xTg mice had normal MWM performance and even better memory retention, which are quite a contrast of the basic requisite of cognitive deficits in an AD mouse model. While most AD mouse models that have been tested in the MWM and other memory tests show AD related cognitive deficits, there appears to be inconsistencies in the manifestation and timing of those cognitive deficits [66, 67]. Notably, both impaired and enhanced fear conditioning memory have been noted in AD mice [68]. Furthermore, the emergence and progression of cognitive deficits can vary between and within AD models [66, 67, 69, 70]. For example, cognitive deficits noted in young 3xTg-AD mice were no longer apparent in middle aged mice [71]. This disappearance of cognitive deficits is consistent with the reduced or lack of spatial and non-spatial cognitive impairments also noted in 14+ month old 3xTg mice [72]. Thus, our results are consistent with the heterogeneity and, at times, paradoxical observations in commonly used AD mouse models. These inconsistencies are likely related to differences in age, types of conducted test, sensitivities and dynamic ranges of behavioral paradigms, and in the ability to parse out age-independent behavioral dysfunction from age-dependent cognitive

abnormalities. Evidently, further studies are needed to gain critical insights into the behaviorally relevant changes occurring during the development of AD pathology and how anesthetics impinge on those behavioral changes. Nonetheless, our neuropathology data are in line with the spatial learning and memory impairments, but some of these observations are in contrast to several other studies in AD and NonTg mice [10, 11, 73].

## Isoflurane impaired synaptic transmission and short term plasticity

Basal synaptic transmission was hyper-excitable in the old 3xTg mice, in agreement with previous reports in various other AD models [71, 74–78]. Nonetheless, repeated Iso exposures significantly depressed basal synaptic transmission in old 3xTg mice and in both young and old mice NonTg mice as previously noted for many excitatory synapses [79–81]. Given that hyper-excitability in APP and J20 AD models is believed to be associated with pathological dysfunctions in interneurons in cortical and DG circuits [74, 82], the noted Iso effects are highly suggestive of a direct action on GABAergic inhibition at CA1 synapses. It remains to be seen whether our treatment paradigm leads to long-term increase in spontaneous or evoked inhibitory synaptic activities at the single cell level.

Another possible explanation for the depression of excitatory synaptic transmission is a reduction in the probability of neurotransmitter release, a form of short term plasticity (STP). STP is induced by $Ca^{2+}$ accumulating in presynaptic nerve terminals during repetitive action potentials and serves as an important mechanism for modifying neural circuits during computation [83]. We noted that sham-treated old 3xTg mice displayed less facilitation compared to age-matched NonTg mice. This observation is consistent with previous reports in which acute application of Aβ protein or the progression of Aβ pathology significantly affects excitatory neurotransmitter release probability, thereby reducing synaptic facilitation at major hippocampal synapses [74, 84]. However, the noted hyper-excitability in old 3xTg mice is difficult to connect with the less facilitating synapses in these old mice, but it appears to be a common observation in APP models [76, 85].

## Altered synaptic plasticity in Alzheimer's disease

There have been no studies so far examining whether anesthetic exposure affects the synaptic capacity for LTP in the hippocampus of 3xTg mice. Here, we show that repeated Iso depressed or prevented LTP in the CA1 hippocampus of neonatal 3xTg and NonTg mice. This level of impairment was similarly noted in old NonTg mice, an observation that contrasts with the improved LTP noted in 4–5 months old mice 24 h after a single exposure to Iso [54]. These discrepant results are likely due to differences in age and exposure paradigms. In contrast to our noted LTP impairments in NonTg mice, LTP in the 3xTg mice was rather normalized by Iso compared to the disturbances noted in sham-treated old 3xTg. This observation is much unexpected given that synaptic plasticity studies generally show that Aβ protein reduces long-term potentiation and/or causes synaptic depression [84, 86–88]. The hyper-excitability in the old mice could explain the lack of effects. Certainly, other mechanisms cannot be overlooked given the many targets of anesthetics. For example, activation of nAChRs localized on hippocampal GABAergic interneurons has been implicated in the mechanism of Iso-mediated inhibition of LTP induction [89, 90].

LTP is a widely accepted model that links synaptic plasticity with memory and this correlation is evident in the neonatal mice as the mediated impairments in LTP appeared to match their cognitive performance. However, we noted a paradoxical enhancement in behavioral performance with impaired LTP in Iso-treated old nonTg mice, a phenomenon that has been noted in previous studies [91–94]. The lack of correlation in LTP and cognition might be

explained by several factors, including induction protocols and age-dependent alterations in the induction of LTP [93, 95]. Further, in addition to its association with decline in cognition, aging is associated with a shift in synaptic plasticity favoring NMDA-independent LTP over the NMDA-dependent form of LTP [95]. Interestingly, behavioral performance appears to correlate more strongly with NMDAR-LTP in young rats, whereas NMDAR-independent LTP correlates with behavioral performance mostly in aged rats [62]. Because the LTP protocol used in our study did not isolate NMDAR-LTP or NMDAR-independent LTP, we cannot rule out the possibility that it favored the induction of an LTP mechanism selectively sensitive to Iso and lacking of any association with SC-CA1 synaptic pathways responsible for some spatial cognitive functions in the aged NonTg mice. Further correlative behavioral and electrophysio-logical studies of mice at different ages using diverse LTP induction protocols are needed to determine how repeated anesthetic exposures impinge on the complex relationship between cognition and synaptic plasticity.

## Age-range anesthetic sensitivity

GAs at clinically relevant concentrations and durations induced widespread and significant neurodegeneration dose-dependently in the developing brains of various animal species[96–100], while they rarely caused significant neurodegeneration in the adult or aged brains[101]. The results in the current study are consistent with the findings in the literature in regard to the correlation of behavioral abnormalities with widespread neurodegeneration in early post-natal brain [9, 30, 96, 99, 102] and the commonly noted behavioral as well as synaptic distur-bances in the old mice in the absence of any measureable neurodegeneration [9, 10, 20, 22]. Although the underlying mechanisms for these age-range sensitivity differences are not clear, the developmental state of the brain during anesthetic exposures could be a contributing fac-tor. Exposures of the neonatal brain occur during the critical period of brain development, a period characterized by programmed cell death, pruning of exuberant synaptic connections, stabilization of remaining connections, and changes to the GABA current reversal potential. These dynamic processes may have rendered the early postnatal brains more susceptible to cell death, leading to the noted behavioral and synaptic alterations described in our results. According to this posit, the old brains would not be expected to exhibit cell death, but aging processes such as reduced oxidative phosphorylation and increase of free radicals amongst others could have rendered neural circuits vulnerable to repeated anesthetic exposures, leading behavioral and synaptic alterations in the absence of cell death as described in our results. Thus, our results in regard to age-range sensitivity are consistent with the literature and sug-gest that the mechanisms by which anesthetics impinge on brain functions in neonates and old mice may depend on the inherent differences in the vulnerability of their neural circuits to insults during the developmental and aging processes.

## Limitation

In spite of the significance of the results of our study, we have noted several small caveats that should be taken into account in regard to their clinical relevance. First, although onset of AD can be at young age in familiar AD (FAD), it only composed of less than 5% of all AD patients. Second, we elected to use a treatment paradigm in which mice were exposed to Iso for 2 hours over 5 consecutive days because it has been shown to be well tolerated by our transgenic mice while inducing robust neurotoxicity and behavioral abnormalities [20]. However, the clinical relevance of such exposure paradigm is uncertain given that patients are unlikely to undergo surgical procedures in 5 consecutive days. Nonetheless, the exposure paradigm and findings of this study have provided significant insights into the detrimental impact of repeated use of

anesthetics in clinical settings. Thirdly, while the use of Iso in this study is consistent with majority of studies on anesthetic-mediated neurotoxicity studies in rodents, sevoflurane should be considered in future studies as it has become the preferred GA in clinical settings. This is particularly important given the ambiguity in comparative strength of neurotoxicity *in vivo*, with one study reporting that equipotent doses of sevoflurane and Iso can to induce neurotoxicity to similar degree in rodents [103], while another suggests that sevoflurane causes less neurotoxicity than Iso [104]. It is plausible to assume that sevoflurane will have similar effects as isoflurane but with less potency in this study. Overall, this study is limited for its importance as a true translational study. Finally, we elected not to measure possible respiratory and metabolic changes during our repeated Iso exposures based on the lack of changes in those parameters in one of our previous studies [104] as well as one from another group [23] using similar exposure conditions, but Johnson and colleagues [105] recently reported that Iso delivered in slightly different conditions from ours resulted in respiratory and metabolic changes in early postnatal and adult mice after 2 hours of Iso exposure using more detailed measurements [105] than Liang [104] and Xie [23] previously described. Although the noted metabolic changes and the Iso exposures did not lead to significant neurotoxicity and cognitive dysfunctions after 2 hours, it might be more prudent to correlate all future anesthetic-mediated neurotoxicity studies, even those of relative short durations, with detailed respiratory and metabolic measurements as described by Johnson and colleagues [105].

## Conclusion

In summary, we provide strong evidence that repeated Iso exposures exacerbated neuropathology, impaired cognition and LTP in 3xTg mice in an age-dependent manner. Given that recent advances in diagnostic, therapeutic, and surgical procedures have significantly increased the possibilities of repeated interventions requiring multiple exposures to anesthetics, our study has provided significant insights into the impact of repetitive use of anesthetics on cognition in clinical settings.

## Supporting information

**S1 Fig. Neonates uncut Western blots.** Representative blots of cleaved caspase 9 (A, 40kDa), caspase 12 (B, 42kDa), Bax (C, 20kDa), BCl-2 (D, 28kDa), and b-actin (E & F, 42kDa. Upper bands indicated in blots A & B by 50 (Caspase 9) and 55 kDa (caspase 12) are procaspase 9 and uncleaved caspase 12, respectively.
(TIF)

**S2 Fig. Old mice uncut Western blots.** Representative blots of cleaved caspase 9 (A, 40kDa), caspase 12 (B, 42kDa), Bax (C, 20kDa), BCl-2 (D, 28kDa), and b-actin (E & F, 42kDa. Upper bands indicated in blots A & B by 50 (Caspase 9) and 55 kDa (caspase 12) are procaspase 9 and uncleaved caspase 12, respectively.
(TIF)

## Acknowledgments

We appreciate the valuable discussions and support from Roderic G. Eckenhoff, MD, Maryellen F. Eckenhoff, PhD, and Lee A. Fleisher, MD at the Department of Anesthesiology and Critical Care, University of Pennsylvania Perelman School of Medicine, Philadelphia, PA 19104, USA.

## Author Contributions

**Conceptualization:** Huafeng Wei.

**Data curation:** Donald J. Joseph, Chunxia Liu, Jun Peng, Ge Liang.

**Formal analysis:** Donald J. Joseph, Ge Liang.

**Funding acquisition:** Huafeng Wei.

**Methodology:** Donald J. Joseph, Chunxia Liu, Ge Liang.

**Supervision:** Huafeng Wei.

**Validation:** Jun Peng.

**Writing – original draft:** Donald J. Joseph.

**Writing – review & editing:** Chunxia Liu, Ge Liang, Huafeng Wei.

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
