## [Decision Letter · Decision Letter 0]

29 Aug 2019

[EXSCINDED]

PONE-D-19-17531

Isoflurane Mediated Neuropathological and Cognitive Impairments in the Triple Transgenic Alzheimer’s Mouse Model Are Associated with Hippocampal Synaptic Deficits in an Age-dependent Manner

PLOS ONE

Dear Dr. Wei,

Thank you for submitting your manuscript to PLOS ONE. After careful consideration by three Reviewers and an Academic Editor, please make the suggested corrections posed by the Reviewers so I can render a decision on this manuscript.

**Comments to the Author**

1. Is the manuscript technically sound, and do the data support the conclusions?

Reviewer #1: Yes

Reviewer #2: Yes

Reviewer #3: Yes

2. Has the statistical analysis been performed appropriately and rigorously? 

Reviewer #1: Yes

Reviewer #2: Yes

Reviewer #3: Yes

3. Have the authors made all data underlying the findings in their manuscript fully available?

Reviewer #1: Yes

Reviewer #2: Yes

Reviewer #3: Yes

4. Is the manuscript presented in an intelligible fashion and written in standard English?

Reviewer #1: Yes

Reviewer #2: Yes

Reviewer #3: Yes

5. Review Comments to the Author

Reviewer #1: The study by Joseph et al described the effects of isoflurane on pathophysiology in AD mice and controls. The results show isoflurane exposure worsens AD pathology in a age-dependent manner using a series of endpoints. The study is rigorously conducted and the findings are important. Would be useful to discuss the role of (multiple) surgeries in the context of this study as a limitation.

Reviewer #2: This is an interesting and novel study. It shows that repeated isoflurance exposure induces neuropathological and cognitive changes in the triple transgenic Alzheimer’s mouse model. And these changes are associated with hippocampal synaptic deficits in an age-dependent way. These findings refresh my recognition on the topic of the effect of inhalational anesthetic isoflurane on the brain of aged or neonatal rodents.

Based on the previously studies, however, I have several questions as follow:

1. Sevoflurane is more commonly used in the clinical settings while this research investigated the effect of isoflurane. As reported previously, sevoflurane is likely to be less neurotoxic in the model explored in current study. I wonder whether sevoflurane would induce the similar effect in the model.

2. The authors mentioned that blood gases were not examined in the section of isoflurane exposure paradigm as 1.5% isoflurane did not affect the ABG. However, a recent study found that this isoflurane exposure paradigm could induce significant disturbance in respiratory and metabolic parameters(https://doi.org/10.1371/journal.pone.0213543). In my opinion, the ABG analysis is needed.

3. The repeated exposure is five times. I am curious whether the changes is relevant to the exposure times. In other word, why you choose exposure 5 times, but not three times or two times?

Reviewer #3: This is a nice written manusript to descript important findigs regarding the anesthesia neurotoxicity as well as the underlying mechanisms in both young and old rodents. The studiesare important in the area. The authors may want to futher discuss the obvious difference between young and older after the anesthesia.

6. PLOS authors have the option to publish the peer review history of their article (what does this mean?). If published, this will include your full peer review and any attached files.

**Do you want your identity to be public for this peer review?** For information about this choice, including consent withdrawal, please see our Privacy Policy.

Reviewer #1: No

Reviewer #2: No

Reviewer #3: No

We would appreciate receiving your revised manuscript by February, 2020. To enhance the reproducibility of your results, we recommend that if applicable you deposit your laboratory protocols in protocols.io, where a protocol can be assigned its own identifier (DOI) such that it can be cited independently in the future. For instructions see: http://journals.plos.org/plosone/s/submission-guidelines#loc-laboratory-protocols

We look forward to receiving your revised manuscript.

Kind regards,

Stephen D. Ginsberg, Ph.D.

Section Editor

PLOS ONE
---

## [Author Response · Author response to Decision Letter 0]

13 Sep 2019

Reviewer #1: The study by Joseph et al described the effects of isoflurane on pathophysiology in AD mice and controls. The results show isoflurane exposure worsens AD pathology in an age-dependent manner using a series of endpoints. The study is rigorously conducted, and the findings are important. Would be useful to discuss the role of (multiple) surgeries in the context of this study as a limitation.

Response: This is an excellent point and we have addressed it in the newly added limitations portion of the discussion.

Because the appearance of neurodegeneration and behavioral abnormalities appears to depend on multiple exposures to anesthesia, we elected to use a repeated exposure paradigm of 2hr over 5 consecutive days. In addition, this repeated exposure paradigm has been shown to be well tolerated transgenic mice while inducing robust neurotoxicity and behavioral abnormalities [1]. However, we agree that the clinical relevance of such exposure paradigm is uncertain given that patients are less likely to undergo surgical procedures in 5 consecutive days. Therefore, we emphasize that point as a limitation to the study.

Reviewer #2: This is an interesting and novel study. It shows that repeated isoflurance exposure induces neuropathological and cognitive changes in the triple transgenic Alzheimer’s mouse model. And these changes are associated with hippocampal synaptic deficits in an age-dependent way. These findings refresh my recognition on the topic of the effect of inhalational anesthetic isoflurane on the brain of aged or neonatal rodents. Based on the previously studies, however, I have several questions as follow:

1. Sevoflurane is more commonly used in the clinical settings while this research investigated the effect of isoflurane. As reported previously, sevoflurane is likely to be less neurotoxic in the model explored in the current study. I wonder whether sevoflurane would induce the similar effect in the model.

Response: This is an excellent point and we have addressed it in the newly added limitations portion of the discussion.

While the use of Isoflurane in this study is consistent with majority of studies on anesthetic-mediated neurotoxicity studies on anesthetics in rodents, we agree that sevoflurane should be considered in future studies as it has become the preferred GA in clinical settings. This is particularly important given the ambiguity in comparative strength of neurotoxicity, with some studies reporting that equipotent doses of sevoflurane and isoflurane can to induce neurotoxicity to similar degree in rodents [2], while our lab previously noted less neurotoxicity by sevoflurane when compared to equipotent dose of isoflurane [3]. 

2. The authors mentioned that blood gases were not examined in the section of isoflurane exposure paradigm as 1.5% isoflurane did not affect the ABG. However, a recent study found that this isoflurane exposure paradigm could induce significant disturbance in respiratory and m metabolic parameters(https://doi.org/10.1371/journal.pone.0213543). In my opinion, the ABG analysis is needed.

Response: This is an excellent point and we have addressed it in the newly added limitations portion of the discussion.

As we alluded to in our method section, we elected not to measure possible respiratory and metabolic changes during our repeated Isoflurane exposures based on the lack of changes in those parameters in one of our previous studies [3] as well as from another study [4] in similar exposure conditions, but it has been reported recently that 2-4 hours exposures to isoflurane resulted to respiratory changes in early postnatal and adult mice [5]. Although the noted metabolic changes and the Isoflurane exposures did not lead to significant neurotoxicity and cognitive dysfunctions after 2 hours, we believe it might be more prudent to correlate all future anesthetic-mediated neurotoxicity studies and this point has been made in the limitation portion of our discussion. The detailed measurements made in that paper uncovered possible confounding factors that we did not observe in our exposure conditions [3]. Although, the differences in the exposure conditions between that study and ours could explain the discrepancy in metabolic changes, we agree that future studies should implement the detailed approach of respiratory and metabolic measurements of Johnson and colleagues [5] in their analysis of anesthetic-mediated neurotoxicity.

3. The repeated exposure is five times. I am curious whether the changes are relevant to the exposure times. In other word, why you choose exposure 5 times, but not three times or two times?

Response: This is an excellent point and we have addressed it in the newly added limitations portion of the discussion.

This comment is addressed above based on a question from reviewer 1. Basically, because the appearance of neurodegeneration and behavioral abnormalities appears to depend on multiple exposures to anesthesia, we elected to use a repeated exposure paradigm of 2 hours over 5 consecutive days because it has been shown to be well tolerated by mice and result in robust neurotoxicity and behavioral abnormalities [1]. 

Reviewer #3: This is a nice written manuscript to descript important findings regarding the anesthesia neurotoxicity as well as the underlying mechanisms in both young and old rodents. The studies are important in the area. The authors may want to further discuss the obvious difference between young and older after the anesthesia.

Response: This is an excellent point and we have addressed it in the newly added “Age-range anesthetic sensitivity” portion of the discussion.

GAs at clinically relevant concentrations and durations induced widespread and significant neurodegeneration dose-dependently in the developing brains of various animal species [97-101], while they rarely caused significant neurodegeneration in the adult or aged brains [102]. The results in the current study are consistent with the findings in the literature in regard to the correlation of behavioral abnormalities with widespread neurodegeneration in early postnatal brain [9, 28, 95-97] and the commonly noted behavioral as well as synaptic disturbances in the old mice in the absence of any measureable neurodegeneration [9, 10, 20, 35]. Although the underlying mechanisms for these age-range sensitivity differences are not clear, the developmental state of the brain during anesthetic exposures could be a contributing factor. Exposures of the neonatal brain occur during the critical period of brain development, a period characterized by programmed cell death, pruning of exuberant synaptic connections, stabilization of remaining connections, and changes to the GABA current reversal potential. These dynamic processes may have rendered the early postnatal brains more susceptible to cell death, leading to the noted behavioral and synaptic alterations described in our results. According to this posit, the old brains would not be expected to exhibit cell death, but aging processes such as reduced oxidative phosphorylation and increase of free radicals amongst others could have rendered neural circuits vulnerable to repeated anesthetic exposures, leading behavioral and synaptic alterations in the absence of cell death as described in our results. Thus, our results in regard to age-range sensitivity are consistent with the literature and suggest that the mechanisms by which anesthetics impinge on brain functions in neonates and old mice may depend on the inherent differences in the vulnerability of their neural circuits to insults during the developmental and aging processes.

References

1. Bianchi SL, Tran T, Liu C, Lin S, Li Y, Keller JM, et al. Brain and behavior changes in 12-month-old Tg2576 and nontransgenic mice exposed to anesthetics. Neurobiol Aging. 2008;29(7):1002-10. doi: 10.1016/j.neurobiolaging.2007.02.009. PubMed PMID: 17346857; PubMed Central PMCID: PMC4899817.

2. Istaphanous GK, Howard J, Nan X, Hughes EA, McCann JC, McAuliffe JJ, et al. Comparison of the neuroapoptotic properties of equipotent anesthetic concentrations of desflurane, isoflurane, or sevoflurane in neonatal mice. Anesthesiology. 2011;114(3):578-87. doi: 10.1097/ALN.0b013e3182084a70. PubMed PMID: 21293251.

3. Liang G, Ward C, Peng J, Zhao Y, Huang B, Wei H. Isoflurane causes greater neurodegeneration than an equivalent exposure of sevoflurane in the developing brain of neonatal mice. Anesthesiology. 2010;112(6):1325-34. doi: 10.1097/ALN.0b013e3181d94da5. PubMed PMID: 20460994; PubMed Central PMCID: PMC2877765.

4. Xie Z, Culley DJ, Dong Y, Zhang G, Zhang B, Moir RD, et al. The common inhalation anesthetic isoflurane induces caspase activation and increases amyloid beta-protein level in vivo. Ann Neurol. 2008;64(6):618-27. doi: 10.1002/ana.21548. PubMed PMID: 19006075; PubMed Central PMCID: PMC2612087.

5. Johnson SC, Pan A, Sun GX, Freed A, Stokes JC, Bornstein R, et al. Relevance of experimental paradigms of anesthesia induced neurotoxicity in the mouse. PLoS One. 2019;14(3):e0213543. doi: 10.1371/journal.pone.0213543. PubMed PMID: 30897103; PubMed Central PMCID: PMC6428290.

---

## [Editor Report · Decision Letter 1]

24 Sep 2019

Isoflurane Mediated Neuropathological and Cognitive Impairments in the Triple Transgenic Alzheimer’s Mouse Model Are Associated with Hippocampal Synaptic Deficits in an Age-dependent Manner

PONE-D-19-17531R1

Dear Dr. Wei,

We are pleased to inform you that your manuscript has been judged scientifically suitable for publication and will be formally accepted for publication once it complies with all outstanding technical requirements.

With kind regards,

Stephen D. Ginsberg, Ph.D.

Section Editor

PLOS ONE

---

## [Editor Report · Acceptance letter]

1 Oct 2019

PONE-D-19-17531R1 

Isoflurane Mediated Neuropathological and Cognitive Impairments in the Triple Transgenic Alzheimer’s Mouse Model Are Associated with Hippocampal Synaptic Deficits in an Age-dependent Manner 

Dear Dr. Wei:

I am pleased to inform you that your manuscript has been deemed suitable for publication in PLOS ONE. Congratulations! Your manuscript is now with our production department. 

With kind regards,

on behalf of

Dr. Stephen D Ginsberg 

Section Editor

PLOS ONE